# 3D PCL/Gelatin/Genipin Nanofiber Sponge as Scaffold for Regenerative Medicine

**DOI:** 10.3390/ma14082006

**Published:** 2021-04-16

**Authors:** Markus Merk, Orlando Chirikian, Christian Adlhart

**Affiliations:** 1Institute of Chemistry and Biotechnology, Zurich University of Applied Sciences ZHAW, 8820 Wädenswil, Switzerland; merk@ucsb.edu; 2Biomolecular Science and Engineering, University of California Santa Barbara UCSB, Santa Barbara, CA 93106, USA; ochirikian@ucsb.edu

**Keywords:** self-assembly, 3D electrospun nanofibrous scaffold, nanofiber aerogels, tissue engineering, electrospun sponge, polycaprolactone, biodegradation

## Abstract

Recent advancements in tissue engineering and material science have radically improved in vitro culturing platforms to more accurately replicate human tissue. However, the transition to clinical relevance has been slow in part due to the lack of biologically compatible/relevant materials. In the present study, we marry the commonly used two-dimensional (2D) technique of electrospinning and a self-assembly process to construct easily reproducible, highly porous, three-dimensional (3D) nanofiber scaffolds for various tissue engineering applications. Specimens from biologically relevant polymers polycaprolactone (PCL) and gelatin were chemically cross-linked using the naturally occurring cross-linker genipin. Potential cytotoxic effects of the scaffolds were analyzed by culturing human dermal fibroblasts (HDF) up to 23 days. The 3D PCL/gelatin/genipin scaffolds produced here resemble the complex nanofibrous architecture found in naturally occurring extracellular matrix (ECM) and exhibit physiologically relevant mechanical properties as well as excellent cell cytocompatibility. Samples cross-linked with 0.5% genipin demonstrated the highest metabolic activity and proliferation rates for HDF. Scanning electron microscopy (SEM) images indicated excellent cell adhesion and the characteristic morphological features of fibroblasts in all tested samples. The three-dimensional (3D) PCL/gelatin/genipin scaffolds produced here show great potential for various 3D tissue-engineering applications such as ex vivo cell culturing platforms, wound healing, or tissue replacement.

## 1. Introduction

Electrospun nanofibers have become a commonly used material for the fabrication of artificial scaffolds in regenerative medicine [1]. They provide a promising framework for the reconstruction and/or regeneration of various tissues types. These highly versatile nanofiber networks can be produced from a variety of natural [2,3,4,5,6,7] or synthetic [8,9,10,11,12,13] polymers, many of which are extremely tunable in terms of molecular weight, degradation rate, and various rheological properties [14]. Limitations of using natural polymers or decellularized scaffolds alone have been highlighted, suggesting that the mechanical durability and/or stability of the scaffolds is lost when transplanted in vivo. This prohibits the use of such scaffolds to applications where structural integrity is important, as addressed here [15]. More recent work features the value of using natural and synthetic polymers in concert to obtain optimal structure, mechanical and biochemical signals, and bioresorbability properties that natural polymer scaffolds alone are not capable of providing [16]. Tissue engineering for regenerative purposes has come a long way, providing scaffolds for long-term drug delivery and numerous tissue (partial) repair or replacement applications. Tissues that have been targeted include skin [17], bone [6], cartilage [9,18], vascular [19,20], neural [4,21], or cardiac [22]. Several reviews have highlighted the use of nanofiber-based scaffolds and give a broad overview on current production processes, characterization techniques, and potential applications [1,23,24,25,26,27,28]. Such composite materials show great potential in mimicking the naturally occurring extracellular matrix (ECM) by providing optimal structural, chemical, and mechanical stimuli for cellular homeostasis [20,29]. Furthermore, electrospun nanofiber scaffolds have the advantage of being cost-effective and easily scalable [25,30]. However, most nanofiber scaffolds are produced in the form of two-dimensional (2D) nanofiber mats [31]. In order to overcome the limitations associated with a 2D culturing system (i.e., limited nutrient diffusion, decreased cell penetration), new approaches must be identified [20,32]. A possible solution is the production of highly porous three-dimensional (3D) nanofiber sponges or aerogels from short nanofibrous building blocks using a self-assembly process in combination with subsequent freeze drying and cross-linking steps [33,34,35,36,37,38,39,40] or the combination of electrospinning and 3D printing [41]. Recently, several approaches of using preformed nanofibers to assemble 3D nanofiber scaffolds for tissue engineering, in particular for bone, have been reported [42,43,44,45,46,47,48,49,50,51]. This is in contrast to alternative approaches for porous 3D scaffolds, e.g., using sol–gel processes [52].

Chen et al. were the first to exhibit the ability of forming biologically relevant 3D assembled nanofiber scaffolds by using biodegradable poly(lactic acid) (PLA)/gelatine polymer blends [32]. However, the use of glutaraldehyde (GA) as a cross-linking agent may have potential toxic effects [53], compromising its utility in vivo [24,54]. For this reason, naturally occurring cross-linking agents with low toxicity such as genipin have become increasingly popular [55]. Genipin, which is found in the fruits of Gardenia jasminoides Ellis, is about 5000–10,000 times less cytotoxic than GA and serves as a suitable replacement in terms of cross-linking capability [55,56,57]. Therefore, we focused on combining genipin with well characterized biomimetic materials such as polycaprolactone (PCL) and gelatin to promote cellular adhesion and growth in a 3D environment [6,11,58,59].

An ideal biomaterial should be progressively degradable and bioresorbable to support the replacement of normal tissue without causing a chronic foreign body reaction [30,60,61]. It should also provide a mechanical and biochemical environment that mimics the native tissue [60]. Biomechanical properties such as the elastic modulus (Young’s modulus) vary significantly between organs and tissues and can range from a few hundred pascals to thousands of kilopascals [62]. A large subset of cellular processes including proliferation, migration, and survival is impacted by the mechanical properties of the ECM on which, or within which, cells are being cultured [63,64]. Indeed, studies on myoblasts grown on substrates with Young’s moduli comparable to mature muscle tissues (≈12 kPa) showed improved muscle differentiation compared to cells grown on either softer or stiffer materials [65]. Therefore, it is imperative to choose the appropriate biomaterial based on the environment it is replacing.

Here, we demonstrate the production of highly porous nanofiber sponges with physiologically relevant mechanical properties from PCL/gelatin/genipin nanofiber building blocks. The involved manufacturing process is flexible in terms of shape and rheological properties, and it allows overcoming known limitations of close-packed fiber sheets. The favorable fibrous microenvironment of the material is kept intact, and its potential use in regenerative medicine is shown using human dermal fibroblasts (HDF).

## 2. Materials and Methods

### 2.1. Preparation of Electrospun PCL/Gelatin Nanofibers

Two-dimensional (2D) electrospun PCL/gelatin nanofiber mats were prepared as described previously [31,66]. Briefly, PCL (molecular weight 80,000, Sigma-Aldrich, Buchs, Switzerland) and gelatin (from cold water fish skin, Sigma-Aldrich, Buchs, Switzerland) were dissolved together in hexafluoro-2-propanol (Sigma-Aldrich, Buchs, Switzerland) at a concentration of 10% (*w*/*v*—i.e., a mass concentration of *γ* = 100 g L^−1^) with sufficient stirring for 12 h at room temperature. Before electrospinning, the solution with a mass ratio of 1:2 (gelatin to PCL) was thoroughly mixed to avoid phase separation. The homogeneous solution was electrospun at an applied voltage of 13 kV, a flow rate of 0.3 mL h^−1^, and at a distance of 15 cm.

### 2.2. Fabrication of 3D PCL/Gelatin/Gelatin Scaffolds Cross-Linked with Genipin

Three-dimensional (3D) scaffolds were prepared by the following five steps: (1) electrospun PCL/gelatin nanofiber mats were cut into small pieces (<1 cm^2^) and suspended in cold deionized water (dH_2_O); (2) the suspended nanofiber mat pieces were homogenized with a mixer B-400 (Büchi, Flawil, Switzerland) for 30 cycles under liquid nitrogen cooling; (3) the uniform nanofiber dispersions were poured into a metal mold and frozen at −20 °C for 1 h followed by freeze-drying using a LyoPro 3000 (Heto, Thermo Fischer Scientific, Waltham, MA, USA) for 24 h, at −53 °C, 0.5 mbar; (4) 3D scaffolds were immersed in a 10 mL ethanol (absolut, Sigma-Aldrich, Buchs, Switzerland) solution containing either 2.5%, 1.0% or 0.5% (*w*/*v*) genipin (98%, Fluorochem, Hadfield, Derbyshire, UK) and incubated for 24 h at room temperature; (5) the resulting cross-linked scaffolds were washed with ethanol and dH_2_O and either stored at 4 °C in dH2O or freeze-dried for 24 h and stored under vacuum.

### 2.3. Fourier Transform Infrared Spectroscopy

Attenuated total reflection Fourier transform infrared spectroscopy (ATR-FTIR) analysis of the scaffolds as well as the reference materials were performed using a Tensor37 FTIR spectrometer (Bruker, Karlsruhe, Germany) equipped with a diamond ATR MkII sampling cell (Specac Golden Gate, Orpington, Kent, UK). Spectra were recorded in a wavenumber range of 4000–6000 cm^−1^, with a resolution of 4 cm^−1^, and with a total of 10 scans per sample.

### 2.4. Differential Scanning Calorimetry

The scaffolds were investigated with differential scanning calorimetry (DSC) to study their thermal transitions and the chemical cross-linking reaction. The cooling and heating rates of the DSC (Mettler Toledo, Greifensee, Switzerland) were maintained at 10 °C min^−1^ within a dynamic temperature range of 24–100 °C. Measurements were made using a 40 µL standard alumina crucible with an average sample size of 0.3 mg. DSC graphs (average scans of three replicas of the first heating scan) were processed using the MATLAB software (MathWorks, Natick, MA, USA).

### 2.5. Determination of Cross-Linking Degree

In order to determine the cross-linking degree of the scaffolds a 2,4,6-trinitrobenzene sulfonic acid (TNBS) assay was performed as previously described [33]. Briefly, between 1.5 mg and 2.0 mg of scaffold were initially immersed in 1 mL sodium hydrogen carbonate (4% (*w*/*v*)) (Sigma-Aldrich, Buchs, Switzerland) and then in 0.5 mL TNBS solution (1% (*w*/*v*)) (Sigma-Aldrich, Buchs, Switzerland) in dH_2_O. Subsequently, the scaffolds were incubated at 41 °C and 750 rpm using a thermoshacker (Allsheng instruments, Hangzhou, Zhejiang, China). Then, 2 mL of 6 mol L^−1^ HCl (Sigma-Aldrich, Buchs, Switzerland) were added, and the scaffolds were incubated again at 70 °C for 1 h and 45 min. Then, 5 mL of dH_2_O were added to the resulting clear yellow supernatant, and the optical absorbance of the solution was measured using an UV-Vis spectrophotometer SPECORD S 600 (Analytik Jena GmbH, Jena, Germany) at a wavelength of 349 nm. The cross-linking degree ($\alpha_{\mathrm{cl}}$) was calculated as the percentage loss of free primary amine groups after the cross-linking reaction using the following equation [31]:(1)$$\alpha_{\mathrm{cl}}=\left(1-\frac{A_{\mathrm{non}\text{-}\mathrm{cl}}{/m}_{\mathrm{non}\text{-}\mathrm{cl}}}{A_{\mathrm{cl}}{/m}_{\mathrm{cl}}}\right)\times 100\%$$
where $A_{\mathrm{non}\text{-}\mathrm{cl}}$ and $A_{\mathrm{cl}}$ is the absorbance of the sample before and after crosslinking and $m_{\mathrm{cl}}$ and $m_{\mathrm{non}\text{-}\mathrm{cl}}$ refer to the mass of the cross-linked and non-cross-linked sample, respectively.

### 2.6. Morphology, Bulk Density, and Porosity of the 3D Scaffold

The morphology of the nanofibers was observed using a Quanta FEG250 (FEI, Hillsboro, OR, USA) scanning electron microscope (SEM) with acceleration voltage of 5 kV and a secondary electron detector spot size of 3.5. In order to prevent overcharging effects during the measurements, the samples were gold coated for 30 s at 20 mA using a Q150RS sputter (Quorum, Leica Microsystems, Wetzlar, Germany).

The bulk density ($\rho_{\mathrm{bulk}}$) and porosity (*φ*) of the scaffolds were determined using standard approaches for cellular materials according to the following equations [6,37]:(2)$$\rho_{\mathrm{bulk}}=\frac{m}{V}=\frac{4\;m}{\pi d^{2}h}$$
and
(3)$$\varphi =\frac{V_{p}}{V}\times 100\;\%=\frac{\left(V-\frac{m}{\rho_{\mathrm{nf}}}\right)}{V}\times 100\;\%,$$
where *m* is the mass and *V* is the total bulk volume of the scaffold, while *d* and *h* are the diameter and height of the cylindrical scaffold. The pore volume (*V*_p_) and the density of the nanofibers (*ρ*_nf_) was calculated as the mass weighted arithmetic mean of the materials with *ρ*_PCL_ = 1.145 g cm^−3^ and *ρ*_gelatin_ = 1.35 g cm^−3^ [67] (the density of genipin was neglected).

### 2.7. Water Absorption Measurements

The swelling of the cylindrical scaffolds was characterized by immersing pieces with an average diameter of 12 mm and an average mass of 30 mg into dH_2_O using a dynamic tensiometer (DCAT25, DataPhysics, Filderstadt, Germany). The scaffolds were allowed to absorb water for 60 s before being mechanically removed and weighted again for 30 s. The water absorption capacity (*Q*_abs_) was calculated using the following equation:(4)$$Q_{\mathrm{abs}}=\frac{m_{d}-m_{w}}{m_{d}},$$
where *m*_d_ and *m*_w_ are the mass of the dry and the wet scaffolds, respectively. The water absorption velocity was calculated from the mass normalized slope during the initial water absorption phase (a representative linear regression analysis of a scaffold cross-linked with 1% genipin is shown in Appendix A).

### 2.8. Mechanical Tests

Compression stress tests were performed on cylindrical scaffolds with an average height of 17 mm and a diameter of approximately 12 mm using an A.XTplus Texture analyzer (Stable Micro Instruments, Godalming, Surrey, UK). The system was equipped with a flat compression tool (diameter of 25 mm) and calibrated for 100 *N*. The mechanical tests were performed at room temperature under vertical compression with a deformation rate of 1 mm s^−1^ and a maximal compression strain of *ε* = 30. The stress tests under wet conditions were performed by submerging the scaffolds in culture medium (DMEM/F-12, Thermo Fischer Scientific, Waltham, MA, USA) for 30 min and then measuring them as described above while being submerged in the medium. The young’s modulus, *E*, was calculated using the compressive strength, *S*, according to the following equation:(5)$$E=\frac{S}{\varepsilon },$$
where *S* is the maximum load (*F*_max_) divided by the area (*A*) of the compressed sample.

### 2.9. Degradation Study

To characterize the degradation of the PCL/gelatin/genipin scaffolds, we utilized the degradation assay previously described by Felfel et al. [68]. Approximately 10 mg of each sample were placed in a 12-well culture plate containing 3 mL of phosphate-buffered saline (PBS, pH = 7.4 ± 0.2, ThermoFischer Scientific) at 37 °C. At different time points during the 8 weeks of incubation, the specimens were removed, rinsed 3 times with dH_2_O, and dried for 48 h at room temperature under vacuum (Heraeus vacutherm, Hanau, Germany).

Afterwards, the dry weight of the scaffolds was recorded using a micro-scale XA204DR (Mettler Toledo, Greifensee, Switzerland), and the percentage remaining mass was determined using the following equation:(6)$$\mathrm{remaining}\;\mathrm{mass}=\frac{m_{t}}{m_{0}}\times 100\;\%,$$
where *m_t_* is the mass of the degraded sample measured at time *t* after drying and *m*_0_ is the initial mass of the sample at time zero.

### 2.10. Cell Culture

HDF were obtained from CellnTec (Bern, Switzerland). Cells were grown in Dulbecco’s Modified Eagle Medium: nutrient mixture F-12 (DMEM/F-12, Thermo Fischer Scientific, Waltham, MA, USA) with 10% fetal bovine serum (FBS, Sigma-Aldrich, Lot: 065M3352), 1% penicilin–streptomyocin (P/S, Sigma-Aldrich, Buchs, Switzerland), and maintained in a humidified 5% CO_2_ incubator (Binder GmbH, Tuttlingen, Germany) at 37 °C. The cells were kept at a confluence <80% prior to the experiments, and cell passages of *p* = 9 and *p* = 12 were used for the short-term and the long-term study, respectively.

### 2.11. Proliferation of Cells on Scaffolds

Prior to cell seeding, scaffolds were sterilized with 75% ethanol for 1 h, rinsed 3 times with PBS, and incubated overnight in culture medium. On the seeding day, the medium was removed, and HDF were seeded at a density of 5 × 10^4^ cells mL^−1^ on top of the scaffolds. As a positive control, the same number of cells was seeded into a 48-well polystyrene tissue culture plate samples (TCPS) without any scaffold. As a negative control, scaffolds were incubated without cells. The metabolic activity of viable proliferating cells was determined at different time points using the PrestoBlue (PB) assay (Invitrogen) as previously described [6]. Briefly, medium was removed, and PB solution (1:10 in culture medium) was added to the samples and incubated for 1 h at 37 °C. After incubation, 100 µL of the PB solution from each well were transferred to a new black 96-well plate, and the change in fluorescence was measured using a Fluostar Optima (BMG Labtech, Ortenberg, Germany) fluorescence multi-well plate reader with the excitation/emission wavelengths set at 560/590 nm. At the end of the experiment, cell proliferation was confirmed via deoxyribonucleic acid (DNA) quantification using the Quant-IT PicoGreen assay (Molecular Probes, Thermo Fischer Scientific, Waltham, MA, USA) as described by the manufacturer. Briefly, scaffolds were transferred in an Eppendorf tube and mixed with 1 mL of lysis buffer (Tris-HCl 10 mmol L^−1^ pH 7.5, 1 mmol L^−1^ EDTA, 0.1% triton-X, Sigma-Aldrich, Buchs, Switzerland). Then, the samples were frozen at −80 °C and thawed for 10 min in an ultrasonication bath. This freeze–thaw cycle was repeated twice. Subsequently, the scaffolds were removed, and the remaining solution was incubated in the dark for 5 min with 1 mL of reagent (dye-solution in TE buffer). After incubation, 200 µL of each sample were transferred to a black 96-well plate, and the change in fluorescence was measured with the excitation/emission wavelengths set at 485/520 nm.

For morphological analysis, the scaffolds were fixed with formalin solution (10%, Sigma-Aldrich, Buchs, Switzerland), washed 2 times with PBS, and dehydrated using an alcohol gradient. Afterwards, the samples were air dried for 4 h and incubated in a hexamethyldisilazane (Sigma-Aldrich, Buchs, Switzerland) atmosphere at room temperature for 24 h before SEM analysis.

### 2.12. Statistical Analysis

All quantitative experiments were performed in triplicates, and the results are expressed as the mean ± standard deviation (SD). Statistical analysis was performed by one-way analysis of variance (ANOVA) with Tukey post-hoc test using OriginPro 9.1. *p*-values of less than 0.05 were considered as statistically significant.

## 3. Results

### 3.1. Preparation of the 3D PCL/Gelatin/Genipin Scaffolds

Manufacturing of the 3D scaffolds occurred by using an adapted version of our previously reported six step process, as illustrated in Figure 1A [36,37]. The process begins with the formation of electrospun PCL/gelatin nanofiber mats. PCL and gelatin were dissolved together in hexafluoro-2-propanol with a mass ratio of 2:1. This ratio was found to have preferential mechanical and cellular properties compared to other ratios such as 1:4, 1:2, and 4:1 [66]. The non-woven nanofiber mats consisted of large (*x* = 610 ± 180 nm in diameter) and small (*x* = 190 ± 60 nm in diameter) fibrils (Appendix A). The nanofiber mats were cut, suspended in ice-cold distilled water, and homogenized until a uniform nanofiber dispersion was obtained. Subsequent freezing and freeze-drying using a directional solid templating approach [32,33] resulted in the highly porous 3D scaffold (Appendix A). To prevent disintegration of this green body in aqueous media, the contact points between the individual fibers were stabilized through cross-linking.

Therefore, the scaffolds were submerged in an alcoholic solution for 24 h containing different mass ratios of genipin (0.5%, 1.0%, and 2.5%). When the crosslinking reaction was complete, the scaffolds were removed and then rinsed once with ethanol and once with deionized water to eliminate the residual genipin from the newly cross-linked PCL/gelatin/genipin scaffolds. During this step, the color of the cross-linked scaffolds changed from white to different shades of blue, indicating that the cross-linking process occurred (Figure 1B).

### 3.2. Morphology and Density of 3D PCL/Gelatin/Genipin Scaffolds

Cross-sections of each scaffold show distinct morphological changes before and after cross-linking (Figure 1C–J). It is apparent that the 3D scaffolds were made from electrospun nanofiber building blocks, which were fused together to form tissue-like structures displaying randomly distributed pores and walls of densely packed nanofibers. No significant differences were observed between different cross-linking degrees in terms of fiber structure or morphology, indicating that the cross-linking process took place on a molecular level (Figure 1D–F,H–J). However, the non-cross-linked samples showed signs of polymer sheets entangled between fiber bundles (arrows in Figure 1C,G). These sheets seem to be observed in other gelatin scaffolds and are reported to be gelatin traces accumulating at the front of the growing ice crystals during the freeze-casting process [69]. After cross-linking and washing, the polymer sheets disappeared, and the uniformly porous micro-structure remained (Figure 1). Additionally, the scaffolds shrunk during the cross-linking process from around 20 mm (green body, prior to crosslinking) to approximately 15 mm in height (after cross-linking) (Figure 1B).

### 3.3. Effect of Cross-Linker Concentration

With increasing amounts of cross-linker, we observed an increased bulk density of the scaffolds, with ranges between 16 and 22 mg cm^−3^ for the cross-linked samples and 14 mg cm^−3^ for the green body (Table 1). When immersed in water, the scaffolds demonstrated a water-holding capacity between 13.1 and 14.2 g g^−1^ (Table 1). The rate of water uptake, driven by capillary forces between hydrophilic nanofibers, was similar between scaffolds with different cross-linking degrees (22–25 mg g^−1^ s^−1^).

To investigate the effect of different genipin concentrations on the cross-linking degree ($\alpha_{\mathrm{cl}}$), we analyzed the amount of free amino groups using the TNBS assay [31,70]. Figure 2A shows that increasing the genipin concentration from 0.5% to 1% significantly increased the cross-linking degree. However, increasing the concentration above 1% resulted only in a modest increase indicating that a plateau was reached at approximately ($\alpha_{\mathrm{cl}}$ = 75%).

We examined whether increasing the genipin concentration could also enhance the mechanical and thermal stability of our samples and would lead to slower degradation rates. Indeed, increasing the amount of cross-linker resulted in increased compression strength in dry and wet conditions (Table 1, Appendix A).

In addition, the cross-linking process significantly improved protection against degradation during the 8 weeks of incubation in PBS (Figure 2B). While the green body showed only a retaining mass of 41% after 8 weeks, degradation was suppressed through cross-linking. A retaining mass of >92% was found irrespective of the applied genipin concentration between 0.5 and 2.5%. To further characterize the cross-linking process, we measured the influence of thermal stability of the scaffolds using differential scanning calorimetry (DSC). Utilizing temperatures between 25 and 100 °C, we observed temperature shifts of up to 1.4 °C between the melting points, demonstrating an increase in thermal stability of the cross-linked samples (Figure 2C). The temperature increased with the genipin concentration, starting with a melting point of 61.2 °C for the green body and elevating to approximately 62.6 °C for the cross-linked sample (2.5% genipin). Attenuated total reflection Fourier transform infrared spectroscopy (ATR-FTIR) measurements revealed the appearance of two new functional groups when PCL was combined with gelatin at 1543 and 1649 cm^−1^ corresponding to the amide I and amide II vibrations, respectively (Figure 2D) [70]. These amide I and II vibrations are due to protonated amine groups on the gelatin fibers, whereas the peak at 1728 cm^−1^ is due to the stretching vibration of carbonyl groups associated with the ester bonds in PCL and genipin [31,53]. The relatively weak gelatin amid III vibration at 1234 cm^−1^ is hidden by the asymmetric C-O-C stretching of PCL [71]. Although the spectra between the different scaffolds look similar, the intensities of the amide I and II peaks increased slightly, whereas the intensity of the carbonyl bond decreased with increased genipin concentration (Appendix A).

### 3.4. Cell Viability and Morphology on 3D PCL/Gelatin/Genipin Scaffolds

In order to investigate the influence of different cross-linker concentrations on the cell viability, HDF were cultured on PCL/gelatin/genipin scaffolds cross-linked with either 0.5%, 1.0%, or 2.5% genipin (*w*/*v*). Two separate experiments were conducted to analyze the short-term (8 days, Appendix A) as well as the long-term (23 days, Figure 3A) cellular response using the PrestoBlue assay. Samples cross-linked with 0.5% genipin showed a tendency to outperform the scaffolds cross-linked with 1.0% or 2.5% in terms of metabolic activity (Figure 3A) and DNA concentration (Figure 3B). These results were observed in both the long-term as well as the short-term culture study (Appendix A).

Due to the limitations of the PrestoBlue assay, only analyzing metabolic activity and not quantifying total cell number, we further confirmed the results by measuring the amount of DNA in the scaffolds at day 23 and compared them with TCPS. As observed in the metabolic activity assay, the DNA content, measured using the PicoGreen assay, was highest in the scaffolds cross-linked with 0.5% genipin (Figure 3B).

To further characterize the health of the cells and inspect HDF morphology and attachment on the 3D PCL/gelatin/genipin scaffolds, we used SEM imaging of cross-sectioned samples from scaffolds after 23 days of culture (Figure 3C–H). We observed healthy cellular states, as indicated by cell elongation and spreading (characteristic of migration and proliferation) in all cross-linking conditions. At higher magnifications (20k–40k), the formation of filopodia was observed (Circles in Figure 3I–K).

## 4. Discussion

In this study, we aimed to synthesize a highly porous nanofibrous scaffold for tissue engineering. The ideal scaffold should maintain its mechanical strength and 3D structure throughout the initial in vitro seeding process as well as after in vivo implantation [55]. Additionally, the synthetic scaffold should not restrict the secretion of ECM or its remodeling process and should degrade over time to reduce the risk of a foreign body response and graft infection [30]. Several aspects, including choice of polymer, cross-linker, and processing conditions affect the final mechanical and morphological properties of the scaffolds [50]. In order to fabricate the ideal 3D scaffold, it is therefore crucial to investigate cross-linking conditions. In previous approaches, GA had been used to chemically cross-link PCL [72] and gelatin [73]. However, to decrease the potential toxic side effects, GA was replaced with the more benign, naturally occurring cross-linking agent genipin [53]. Here, we evaluated the effect of three different genipin concentrations in the range between 0.5 and 2.5%.

We were able to show that cross-linking the green body with genipin was necessary to preserve the 3D structure in wet conditions but did not impact the overall shape and porous architecture. Cross-linking rendered the green body water stable, and it was possible to determine the water-holding capacity. Notably, the values in Table 1 (13.1 to 14.2 g g^−1^) are significantly lower than the theoretical water-holding capacity values between 44.4 g g^−1^ (2.5% genipin) and 59.8 g g^−1^ (0.5% genipin), indicating that the capillary forces were too weak to fill the larger pores. However, our results are comparable with other sponges or aerogels made from PCL [6], pullulan/PVA [37], or PLA/gelatin [9]. A similar water uptake rate between 22 and 25 mg g^−1^ s^−1^ was found irrespective from the cross-linking degree, indicating that the cross-linking degree did not change the architecture of the porous fiber-based scaffolds.

Evaluation of the cross-linking degree ($\alpha_{\mathrm{cl}}$) using the TNBS assay revealed no significant increase beyond a genipin concentration of 1%, which is in agreement with reports in prior work, showing a plateau in crosslinking degree for 2D PCL/gelatin/genipin nanofiber mats [31,70].

Previous work has shown that the different chain lengths acquired during the cross-linking process significantly influence the mechanical properties and thermal stability of genipin-fixed hydrogels [74]. The range of mechanical properties (Young’s modulus) obtained in this study (7–8 kPa in the wet and 19–23 kPa in the dry sate) mimic stiffness measurements obtained from organs and soft tissues such as kidneys (5–10 kPa), smooth or cardiac muscles (10–15 kPa), and intestines (20–30 kPa) [62,63,64]. Thus, highlighting the great potential of the produced scaffolds in replacing and/or repairing native tissue with the elastic modulus range of 10–30 kPa.

To further evaluate the biological relevance of the 3D scaffolds, we conducted a simple degradation analysis (Figure 2B). As expected, cross-linking prohibited degradation (>92% retaining mass after 8 weeks in PBS); however, there were no significant difference in degradation between samples with different cross-linking degrees. These findings may be explained by the slow degradation rate of PCL, which was reported to be 2–4 years for complete resorption in vivo [75], or the lack of enzymes present in the experimental condition, which would facilitate the hydrolytic cleavage of the ester bonds [68].

Based on our findings, we propose that the increased thermal and mechanical stability originate from the inter fiber cross-linking events between the electrospun PCL/gelatin composite fibers and genipin. However, we cannot exclude the potential influence of intra fiber modifications at this point.

The cross-linking mechanism of genipin is poorly understood, but it has been proposed to be initiated through nucleophilic attack by primary amine groups at the olefinic genipin C-3 atom. This leads to the formation of an intermediate aldehyde via a ring-opening mechanism [55,59,76]. In a second step, the newly formed secondary amine can undergo an intramolecular nucleophilic reaction followed by ring closure to form a single-attached genipin. The carbonyl C can also undergo nucleophilic attack by a neighboring amine group to form a single chain cross-link [59]. Another suggested mechanism for a single chain cross-link involves the replacement of the ester group on the genipin molecule by a secondary amide [55,77]. Both mechanisms are compliant with our FTIR observations, where the gradual formation of C = *N* bonds at the expense of C = O bonds during the formation of PCL–genipin–gelatin cross-links was observed [77]. Under alkaline conditions, genipin can undergo a ring-opening polymerization via an aldol condensation reaction, resulting in a long-range intermolecular multi chain cross-link [74]. Blue pigmentation is caused by side reactions [77]. As reported by Fessel et al. and Bi et al., the degree of observed blue pigmentation is proportional to the amount of cross-linker used [55,78]. Previous studies suggest that the change in coloration is due to the spontaneous reaction of genipin with primary amine groups followed by dehydration and dimerization [55,76,79]. The resulting dimers and trimers are responsible for the distinct blue pigmentation of the scaffolds [55,57,74,77,78].

HDF were successfully cultured on the PCL/gelatin/genipin scaffolds for up to 23 days (Figure 3A). In general, all scaffolds showed good cell cytocompatibility and did not impair cell proliferation. On the contrary, a linear increase in metabolic activity of the HDF was observed in all specimens.

The PicoGreen assay showed the highest metabolic activity for the scaffold cross-linked with 0.5% genipin. A possible reason for the slightly decreased total cell number on scaffolds incubated with 2.5% and 1.0% genipin could be due to pH changes induced by higher genipin concentrations [78]. Although no medium acidification was observed, we cannot exclude confounding effects caused by these pH variations. Another explanation for these results could be the reduction of free amine groups on the scaffolds. Genipin cross-linking occurs preferentially on amino acids such as lysin, asparagine, glutamine, and arginine, some of which are also required for the cellular/ECM interactions through integrin binding [80,81]. The loss of recognition sites could hinder the interactions between the cell and its environment, thus leading to reduced adhesion, migration, and proliferation.

One limitation observed with these scaffolds was the low initial cell seeding density compared to TCPS (Figure 3A and Appendix A, first time points). The high porosity of the scaffolds and the resulting rapid flow of medium through the specimens makes it difficult for cell retention and subsequent attachment, which may have led to the observed effect. Functionalization of the PCL nanofibers [80] or the use of a dedicated perfusion bioreactor [43] could overcome this problem. For example, Cause et al. showed that aminolyse treatment of PCL nanofibers improved cell adhesion compared to non-treated samples. Furthermore, the immobilization of RGD sites and the introduction of -COOH groups on the surface of PCL nanofibers could further enhance cell adhesion properties [80].

Ultimately, after 23 days of continuous culture, the 3D PCL/gelatin/genipin scaffolds reached similar levels of DNA content and metabolic activity compared to the 2D standard tissue culture platforms, here indicated as our positive TCPS (Figure 3A,B).

Morphological inspection of the HDF at the same culture state through SEM revealed the presence of filopodia. Filopodia are thin (0.1–0.3 µm), finger-like structures containing densely packed filamentous (F)-actin [82]. These filaments act as antennae for cells to probe their environment and are imperative for proper cell migration and adhesion [80,82]. These characteristics (cell adhesion, spreading, and proliferation of the HDF) were also observed in our short-term study at day 8 (Appendix A).

Although our scaffolds did not outperform TCPS during the test period, we believe that the physiological relevant elastic modulus together with the topographical contributions by the nanofibers and the introduction of a third dimension in our culture platform to be highly beneficial for cellular hemostasis. Previous work demonstrated that the expression of several structural proteins such as integrin α1 and β2 are upregulated in cells cultured on nanofibers compared to solid surfaces [25]. These findings support the influence of nanofiber topography and mechanical forces in facilitating proper cell migration, cytoskeletal organization, intracellular signaling, and cell fate [66].

Overall, the 3D culture platform described here provides a flexible and powerful scaffold to the field of tissue engineering that can be further modified in hopes of achieving fully functional transplantable 3D nanofiber scaffolds for the use in regenerative medicine.

## 5. Conclusions

A highly porous 3D PCl/gelatin/genipin nanofiber based composite scaffold cross-linked with the naturally occurring cross-linker genipin was successfully fabricated via a simple six-step process. The resulting 3D scaffolds showed porous nanofiber networks with extracellular environments and mechanical properties similar to that of native tissue. Chemical cross-linking was necessary to preserve the structural integrity of the specimens in aqueous environments, as demonstrated by the incubation of scaffold in buffer solution for 8 weeks. Mechanical properties of the scaffolds were tuned by varying the concentration of cross-linker, and physiological elastic moduli were obtained with values mimicking soft tissues and organs (i.e., muscle, etc.). Among all tested cross-linker concentrations (0.5%, 1.0%, and 2.5% genipin), samples cross-linked with 0.5% genipin demonstrated the highest metabolic activity and proliferation rates for HDF. These results were consistent for both short-term (8 days) and the long-term (23 days) cultures. Furthermore, SEM images indicated excellent cell adhesion and the characteristic morphological features of fibroblasts in all tested samples. Thus, the 3D PCL/gelatin/genipin scaffold produced here shows great potential for various tissue-engineering applications such as ex vivo cell culturing platforms, wound healing, or tissue replacement.

## Figures and Tables

**Figure 1 materials-14-02006-f001:**
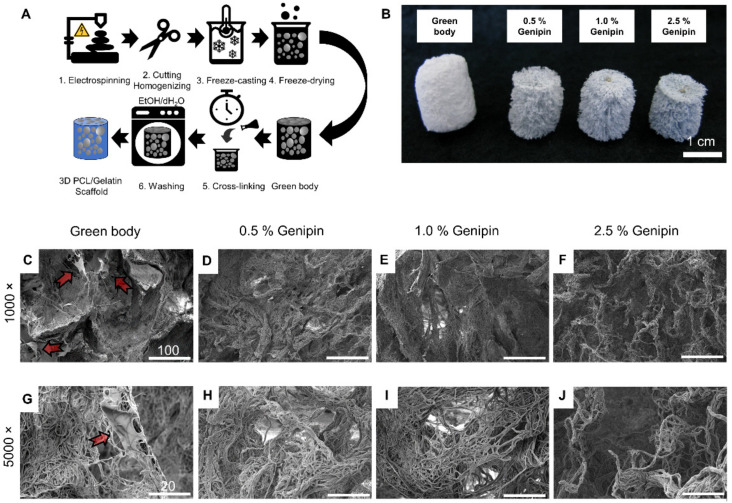
Preparation and morphology of 3D PCL/gelatin scaffolds. (**A**) Schematic illustration of the fabrication process. (**B**) Influence of different cross-linker concentrations on scaffold color and morphology, scale bar = 1 cm. (**C**–**J**) SEM images of cross-sections: (**C**,**G**) scaffolds prior to cross-linking (green body), arrows show polymer sheets in-between the fibers and cross-linked with 0.5% (**D**,**H**), 1.0% (**E**,**I**), and 2.5% genipin (**F**,**J**); (**C**–**F**) scale bar = 100 µm, (**G**–**J**) scale bar = 20 µm.

**Figure 2 materials-14-02006-f002:**
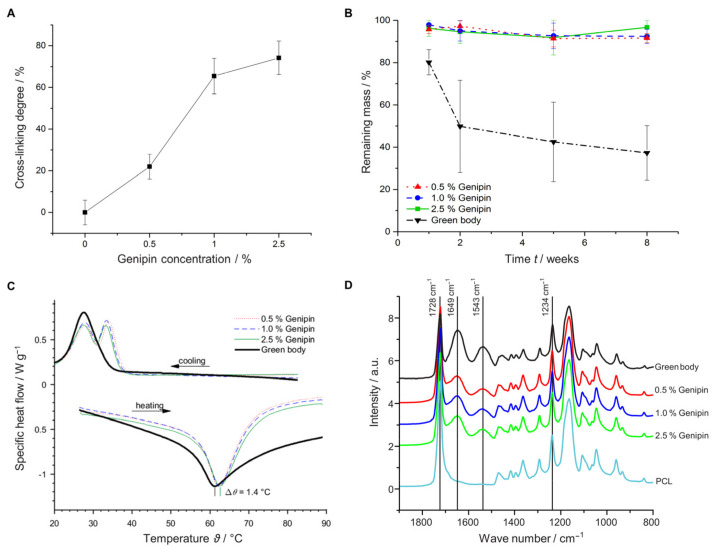
Characterization of scaffolds with different cross-linker concentrations (0.5%, 1.0%, and 2.5% genipin) and the non-cross-linked control (green body). (**A**) Cross-linking degree ($\alpha_{\mathrm{cl}}$). (**B**) Degradation study in buffer solution for 8 weeks. (**C**) DSC measurements. (**D**) ATR-FTIR analysis.

**Figure 3 materials-14-02006-f003:**
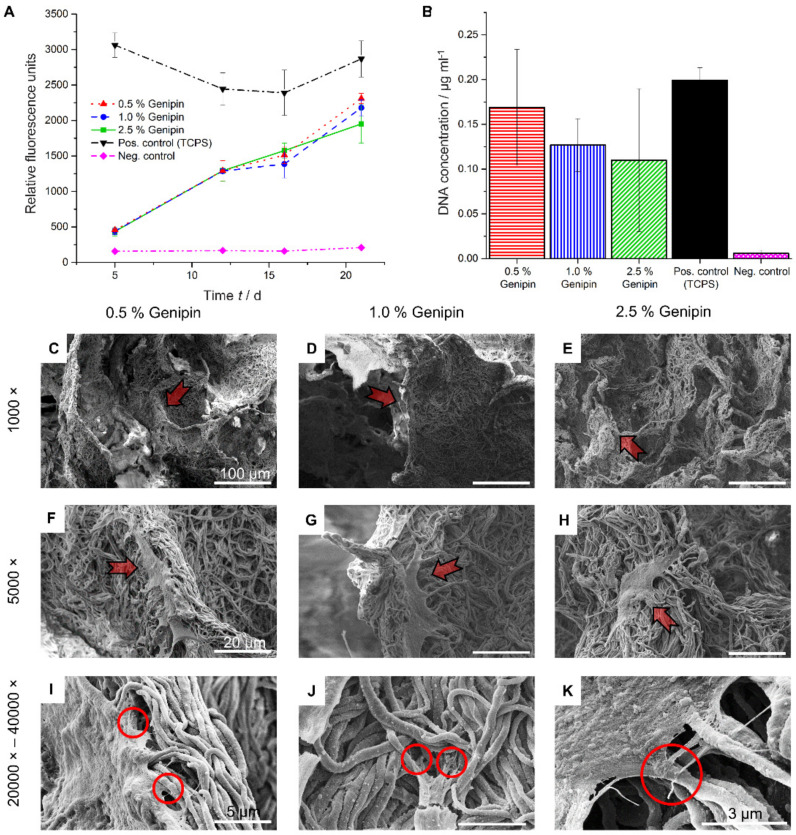
Cell viability and cell morphology on scaffolds. (**A**) Metabolic activity measurements of HDF at various time points using the PrestoBlue assay. (**B**) DNA quantification at day 23 using the PicoGreen assay. (**C**–**K**) SEM images off the cell morphology at the end of the incubation time (day 23), arrows indicate adhering HDF. (**C**,**F**,**I**) Scaffolds cross-linked with 0.5% genipin. (**D**,**G**,**J**) Scaffolds cross-linked with 1.0% genipin. (**E**,**H**,**K**) Scaffolds cross-linked with 2.5% genipin. (**C**–**E**) Scale bar = 100 µm, (**F**–**H**) scale bar = 20 µm, (**I**,**J**) scale bar = 5 µm, (**K**) scale bar = 3 µm.

**Table 1 materials-14-02006-t001:** Physical and mechanical properties for scaffolds with different cross-linker concentration and before cross-linking (green body): bulk density, $\rho_{\mathrm{bulk}}$, porosity, φ, Young’s modulus under dry conditions, $E_{\mathrm{dry}}$, and in culture medium, $E_{\mathrm{wet}}$, water absorption capacity, $Q_{\mathrm{abs}}$, and water absorption rate, ${\dot{Q}}_{\mathrm{abs}}$.

3D Scaffold	$\rho_{\mathrm{bulk}}$/mg cm^−3^	φ/%	$E_{\mathrm{dry}}/\mathrm{kPa}$	$E_{\mathrm{wet}}/\mathrm{kPa}$	$Q_{\mathrm{abs}}$/g g^−1^	${\dot{Q}}_{\mathrm{abs}}$/10^−3^ g g^−1^ s^−1^
Green body	14.3	98.8	10.6 ± 0.27	-	-	-
0.5% Genipin	16.5	98.6	18.8 ± 0.28	6.5 ± 0.17	14.0 ± 1.4	23 ± 2.3
1.0% Genipin	20.2	98.3	22.0 ± 0.76	6.7 ± 0.22	13.1 ± 1.5	25 ± 9.4
2.5% Genipin	22.1	98.2	23.4 ± 0.96	7.8 ± 0.40	14.2 ± 0.3	22 ± 9.1

## Data Availability

The data presented in this study are available on request from the corresponding author.

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
