# Peer review of "3D PCL/Gelatin/Genipin Nanofiber Sponge as Scaffold for Regenerative Medicine"

_materials, 2021, doi:10.3390/ma14082006_

Round 1

Reviewer 1 Report

In the manuscript entitled “3D PCL/Gelatin/Genipin Nanofiber Sponge as Scaffold for Regenerative Medicinence” authors described study on preparation  scaffold for for regenerative medicinence. Obtained 3D scaffolds showed porous nanofiber networks with extracellular environments and mechanical properties similar to that of native tissue.

Below are some questions/suggestions I have for the authors as well as some errors found that should be addressed.

  1. In the introduction I’m missing reference and discussion of major work on 3D scaffold for regenerative medicine, no reference to this substantial contribution to this area is given but furthermore, that research should be discussed in these context as it is highly relevant, for example: Macromolecular Bioscience 2010, 10, 91–100; Micron 2018, 119, 64–71, but many others are available.
  1. DSC: there is no information on whether the second heating was used.
  2. “natriumhydrogencarbonate” is not correct IUPAC chemical name.
  3. Page 3 line 134, there are empty parentheses “The bulk density ( ) and porosity ( )”
  4. Figure 2d ATR‐FTIR spectra, the scale is in reverse order. The unit is “cm-1” so 800 cm-1 is greater than 1,800 cm-1. Figure 2 should be corrected in accordance with the general accepted standard.
  5. Amides has 3 type of vibrations, in the manuscript authors are discussing only amide I and II vibrations.

Author Response

Q: In the introduction I’m missing reference and discussion of major work on 3D scaffold for regenerative medicine, no reference to this substantial contribution to this area is given but furthermore, that research should be discussed in these context as it is highly relevant, for example: Macromolecular Bioscience 2010, 10, 91–100; Micron 2018, 119, 64–71, but many others are available.

A: The article “Macromolecular Bioscience 2010, 10, 91–100” is highly relevant for our work and had already been cited as ref (58). But the applied electrospun PCL nanofibers were an essentially 2D material – we have added it as an example of 2D electrospun materials in the introduction. The article “Micron 2018, 119, 64–71” is relevant since it is showing an alternative sol-gel approach to obtain 3D porous scaffolds. The following was added to the introduction “This is in contrast to alternative approaches for porous scaffolds, e.g. sol gel processes”. Overall, we did not extend the discussion in the introduction to the relevance of 3D scaffolds in general, since there are a number of excellent review articles available and since we want to keep the focus on 3D scaffolds comprised of preformed nanofibers as investigated here.

Q: DSC: there is no information on whether the second heating was used.

A: the first heating was used – the experiment was done with three replicates and their average curves are shown in Figure 2 (information is added to the experimental section)

Q: “natriumhydrogencarbonate” is not correct IUPAC chemical name.

A: corrected to “sodium hydrogen carbonate

Q: Page 3 line 134, there are empty parentheses “The bulk density ( ) and porosity ( )”

A: missing greek symbols rho_bulk and phi were introduced

Q: Figure 2d ATR‐FTIR spectra, the scale is in reverse order. The unit is “cm-1” so 800 cm-1 is greater than 1,800 cm-1. Figure 2 should be corrected in accordance with the general accepted standard.

A: Figure 2D was changed accordingly.

Q: Amides has 3 type of vibrations, in the manuscript authors are discussing only amide I and II vibrations.

A: The peak of the weaker amide III vibration of gelatin at 1240 cm-1 is hidden by the 1234 cm-1 C-O-C asymmetric stretching vibration of PCL. This peak is now marked in Figure 2d and the following sentence was added to the manuscript “The relatively weak gelatin amid III vibration at 1234 cm-1 is hidden by the asymmetric C-O-C stretching of PCL”

Reviewer 2 Report

Please adapt the abstracta s per materials mdpi requirements

You have cited “[1, 20‐22]” however they are little bit old refers, I suggest having a look also to https://doi.org/10.3390/nano10102019

The introduction is quite interesting however our contribution is not very well defined in respect to literature data

Please check this “The bulk density ( ) and porosity ( )”

The methods are well described apart considering some minor typo

“OriginPro” which version ?

From line 293-300 you have used different font size why ?

I don’t have understand very well why 23day was maximum ? why not longer/shorter ?

Please define all the acronyms before their first appearance in text, e.g SEM

Author Response

Q: Please adapt the abstract as per materials mdpi requirements

A: The abstract format and length was adapted accordingly to follow the requirements of “materials”.

Q: You have cited “[1, 20‐22]” however they are little bit old refers, I suggest having a look also to https://doi.org/10.3390/nano10102019

A: We have added the given reference to the introduction and we have added several recent publications using electrospun nanofibers to construct 3D scaffolds for bone tissue regeneration, 10.1002/adhm.201701415, 10.1002/adfm.202005531, 10.3390/nano10102019, 10.1186/s40580-019-0209-y and 10.3390/ijms21010099. Others, such as 10.1016/j.jcis.2018.09.071 or 10.1016/j.cej.2019.03.091 have already been cited in to original submission.

Q: The introduction is quite interesting however our contribution is not very well defined in respect to literature data.

A: Please see the revised document Lines 32-44 and Lines 89-92.

Q: Please check this “The bulk density ( ) and porosity ( )”

A: missing greek symbols rho_bulk and phi were introduced

Q: The methods are well described apart considering some minor typo

A: The manuscript was proof-read by an English native speaker and minor typos were removed

Q: “OriginPro” which version?

A: Version 9.1 was used. This is added in the manuscript

Q: From line 293-300 you have used different font size why ?

A: We apologize for this error and have reviewed the entire document for any formatting errors.

Q: I don’t have understand very well why 23day was maximum ? why not longer/shorter ?

A: 21 to 30 days are a typical end points used for the cultivation of HDF cells to demonstrate successful “long-term” culturing capabilities (e.g. 10.1089/ten.tea.2013.0640, 10.1089/ten.tea.2014.0443, 10.3390/ijms21249596, or 10.1089/ten.teb.2015.0567).

Q: Please define all the acronyms before their first appearance in text, e.g SEM

A: Definitions of the following acronyms were added: polycaprolactone (PCL), Scanning electron microscopy (SEM), two-dimensional (2D), three-dimensional (3D), deoxyribonucleic acid (DNA) and analysis of variance (ANOVA)

Reviewer 3 Report

Dear Authors,

The present study "3D PCL/Gelatin/Genipin Nanofiber Sponge as Scaffold for Regenerative Medicine" aims to investigate the use of Genipin in the cross-linking process of biomaterials. 

It is a very original article and I have phew comments.

the methods are clearly explained but you need to do some corrections in order to cite the products and brands in the correct way.

line 82: "molecular weight 80ʹ000 Sigma‐Aldrich" add St. Louis, Missouri, US)

line 83:Sigma‐Aldrich add St. Louis, Missouri, US)

Line 92: (Büchi......)add  brand, city, and state

line 102,103,108,111  add brand, city and state

line 178,182,196,199,293,287,307,319, add  brand, city and state

Author Response

Q: line 82: "molecular weight 80ʹ000 Sigma‐Aldrich" add St. Louis, Missouri, US)

A: Buchs, CH added (in our specific case)

Q: line 83:Sigma‐Aldrich add St. Louis, Missouri, US)

A: city and country only added at first time use

Q: Line 92: (Büchi......)add  brand, city, and state

A: Flawil, CH added

Q: line 102,103,108,111  add brand, city and state

A: done

Q: line 178,182,196,199,293,287,307,319, add  brand, city and state

A: done

Round 2

Reviewer 1 Report

The authors improved their manuscript significantly,  it is now suitable for publication and I recommend publishing this manuscript.

Reviewer 2 Report

.